# Recommendations for Standardizing Thorax PET–CT in Non-Human Primates by Recent Experience from Macaque Studies

**DOI:** 10.3390/ani11010204

**Published:** 2021-01-15

**Authors:** Marieke A. Stammes, Jaco Bakker, Richard A. W. Vervenne, Dian G. M. Zijlmans, Leo van Geest, Michel P. M. Vierboom, Jan A. M. Langermans, Frank A. W. Verreck

**Affiliations:** 1Department of Parasitology, Biomedical Primate Research Centre, 2288 GJ Rijswijk, The Netherlands; vervenne@bprc.nl (R.A.W.V.); vierboom@bprc.nl (M.P.M.V.); verreck@bprc.nl (F.A.W.V.); 2Animal Science Department, Biomedical Primate Research Centre, 2288 GJ Rijswijk, The Netherlands; bakker@bprc.nl (J.B.); zijlmans@bprc.nl (D.G.M.Z.); geest@bprc.nl (L.v.G.); langermans@bprc.nl (J.A.M.L.); 3Department Population Health Sciences, Division Animals in Science and Society, Faculty of Veterinary Medicine, Utrecht University, 3584 CL Utrecht, The Netherlands

**Keywords:** PET–CT, fluorodeoxyglucose, non-human primates, macaques, thorax, respiratory infections

## Abstract

**Simple Summary:**

With computed tomography (CT) only anatomical information is obtained, but there is no information about function. Positron emission tomography (PET) only gives functional information about a target organ/tissue by the distribution of a specific contrast agent, a radiotracer. Combining the two imaging techniques provides synergy and they can amplify each other. In clinical practice, this combined PET–CT technique is already broadly used. For monkeys, it is increasingly used, with procedures and protocols mainly based on clinical practice. In this publication, we make recommendations about key steps towards standardization and optimization of non-human primate PET–CT (from study preparation to image interpretation).

**Abstract:**

Despite the possibilities of routine clinical measures and assays on readily accessible bio-samples, it is not always essential in animals to investigate the dynamics of disease longitudinally. In this regard, minimally invasive imaging methods provide powerful tools in preclinical research. They can contribute to the ethical principle of gathering as much relevant information per animal as possible. Besides, with an obvious parallel to clinical diagnostic practice, such imaging platforms are potent and valuable instruments leading to a more refined use of animals from a welfare perspective. Non-human primates comprise highly relevant species for preclinical research to enhance our understanding of disease mechanisms and/or the development of improved prophylactic or therapeutic regimen for various human diseases. In this paper, we describe parameters that critically affect the quality of integrated positron emission tomography and computed tomography (PET–CT) in non-human primates. Lessons learned are exemplified by results from imaging experimental infectious respiratory disease in macaques; specifically tuberculosis, influenza, and SARS-CoV-2 infection. We focus on the thorax and use of 18F-fluorodeoxyglucose as a PET tracer. Recommendations are provided to guide various stages of PET–CT-supported research in non-human primates, from animal selection, scan preparation, and operation, to processing and analysis of imaging data.

## 1. Introduction

Respiratory diseases are major causes of death and disability worldwide, affecting all people, regardless of age [1]. This motivates both basic and translational research efforts on pathogenesis and the development of (improved) therapies. For translational research, it is essential to use a model that closely mimics the human respiratory system. Non-human primates (NHPs) comprise valuable species in biomedical research due their close phylogenetic relationship and great similarity to man in various aspects, including physiology, immunity, and susceptibility to infection and disease. NHPs are anticipated to have great predictive validity for the translation of research findings. The use of imaging platforms can register the development of disease over time and/or determine longitudinally the effect of treatment. Moreover, it is the dynamic of disease or longitudinal treatment effect that can be visualized and/or quantified by using imaging modalities, adding to the scientific output of preclinical research.

The most widely used imaging platform for visualizing the lungs is a chest X-ray. Two-dimensional (2D) chest X-ray recordings are sufficient in elucidating clear impact abnormalities at relatively low radiation dose. In general, however, conventional X-ray is limited by the amount of detail it can provide, which is intrinsic to the dimensional reduction of a three-dimensional (3D) structure into a 2D projection and the loss of information by overlay of structures. Three-dimensional (3D) computed tomography (CT) excels the resolution of X-ray-based radiography. Typically, CT is a very powerful modality for visualizing anatomical details towards the detection of pathology. However, there is still a lack of functional information, which can be circumvented by the use of molecular imaging. A widely applied technique in this regard is positron emission tomography (PET), which is particularly powerful, when combined with CT (PET–CT) as hybrid imaging modality. Together, PET–CT merges anatomical and functional information [2]. For functional molecular imaging the use of a specific contrast agent, a radiotracer, is required to be able to assess a biological process through a dedicated molecular target [3]. As PET uses the rapid radioactive decay of positron-emitting isotopes, such a tracer (e.g., F-18 or C-11) is chemically incorporated into a molecule with a high biological relevance. A widely used tracer molecule is fluorodeoxyglucose (FDG) labeled with F-18. FDG is a glucose analogue that gets trapped in metabolically active cells and its accumulation is correlated with the amount of glucose consumption, which is visualized in the image. Increased glucose consumption is related to enhanced expression of glucose transporters (GLUT) and increased hexokinase activity, which are characteristic for cancerous growth or infection-associated inflammation [4]. Next to ^18^F-FDG, there are a lot of different tra-cers with other isotopes in development, all with their own advantages and disadvantages. Unfortunately, only a limited amount of them is approved by the Food and Drug Administration (FDA) for routine clinical use although more can probably be used with good clinical performance [5]. There might be a good opportunity for NHP research to use and compare some of those newly developed radiopharmaceuticals with each other in the coming years.

In recent years, the use of PET–CT has rapidly increased for both clinical and precli-nical purposes [6]. Despite the close resemblance of human subjects and NHPs [7,8], a one-to-one translation of clinical protocols and procedures from man to NHPs is not feasible. PET–CT of NHPs require specific considerations and settings. Currently, no consistent protocols for thorax PET–CTs in NHPs are available. In the present paper we present an overview of those aspects and provide specific recommendations that are based upon our experience from experimental respiratory infection studies in both rhesus macaques (*Macaca mulatta*) and cynomolgus macaques (*Macaca fascicularis*). Results are presented to support the methodological recommendations, which are structured from the preparation phase including animal selection, to the operational phase of PET–CT imaging and the downstream processing and analysis of data

## 2. Methods

### 2.1. Animal Use and Ethics

Data were obtained as part of tuberculosis, influenza, and COVID-19 studies. The scans were performed from January 2017 to September 2020. In total, around 1000 PET–CT protocols on about 250 animals were performed. Imaging and the associated animal handling were an intrinsic part of each study protocol. All housing and animal care procedures took place at the Biomedical Primate Research Centre (BPRC, Rijswijk, The Netherlands). The BPRC is accredited by the American Association for Accreditation of Laboratory Animal Care (AAALAC) and is compliant with European directive 2010/63/EU as well as the “Standard for Humane Care and Use of Laboratory Animals by Foreign Institutions” provided by the Department of Health and Human Services of the US National Institutes of Health (NIH, identification number A5539-01). Each study was performed in compliance with ethical permit from the central committee of animal experiments and specific study protocols were approved prior to start by BPRC’s institutional animal welfare body.

### 2.2. PET–CT Scanner

The scanner used in these studies is a dedicated NHP PET–CT, the Multiscan Large Field of View Extreme Resolution Research Imager (LFER) 150 PET–CT (Mediso Ltd., Budapest, Hungary) with a gantry bore of 26 cm, a crystal size of 2.28 mm^2^ and an axial field of view (FOV) of 15 cm and transaxial 20 cm. The CT operates at submillimeter resolution in combination with ultralow dose imaging. More scanner characteristics are provided in the publication of Sarnjay et al. [9,10].

Besides the LFER 150, there are a couple of other scanners developed dedicated for large animal imaging like NHPs. The MicroPET P4 and MicroPET FOCUS 220 both proved their value in NHP imaging, but are no longer supported by the manufacturer [11]. Two other scanners are currently available. One was recently developed by MR Solutions, the MRS* PET/CT 220, with a bore size of 29.5 cm [12]. The other is a single PET scanner, the Eplus-260 primate PET developed by the Institute of High Energy Physics, Chinese Academy of Sciences, with a bore diameter of 23.6 cm [7]. 

For the scanners mentioned above, the recommendations described in this article can be directly applied. For institutes in which NHPs are scanned in clinical whole-body imagers, the majority of the considerations and settings regarding animal characteristics, except for body composition, scan acquisition, besides scanning time and activity, reconstruction and data analysis could be applied too.

### 2.3. Scanner Room

The scanner room is integrated in the experimental animal facilities and fully compatible with biosafety level 3 containment conditions. To this end, a buffer zone at negative pressure, is located between the scanner room and the hallway for contained entry. Temperature and air pressure, humidity, and ventilation are all monitored and controlled to optimally maintain the performance of the scanner (Figure 1). 

Inside the room there are mobile lead screens to shield and protect personnel from radiation and radioactivity. Those screens are placed around the scanner or the animal preparation table for shielding during incubation and scanning. An additional smaller mobile lead screen with a small lead glass window and a dedicated syringe holder is available to perform the injection of radioactive PET tracer. 

### 2.4. Animal Housing and Care

All animals, both rhesus macaques (*Macaca mulatta*) and cynomolgus macaques (*Macaca fascicularis*), were born and raised at the BPRC in large naturalistic breeding groups until at least 4 years of age before they were moved into experimental facilities. Animals were always socially housed and were between 4 and 21 years old. 

The monkeys were fed commercial monkey pellets (Ssniff, Soest, Germany), approximately 150 g per monkey per day. Fruits and vegetables were provided on a daily basis, and so was enrichment, both in food and non-food items. Before a PET–CT, the animals were fasted overnight. Water was available *ad libitum*. The animal room temperature was maintained at 20 ± 2 °C and relative humidity 50 ± 10%, with 15 air changes per hour and a light/dark cycle of 12 h with fluorescent lighting.

Animals were trained by positive reinforcement to reduce stress and limit the need for sedation for procedures as much as possible, e.g., voluntary entry of the balcony cage. The home cage included a balcony box, where the macaques received an intramuscular (IM) injection of a combination of ketamine (ketamine hydrochloride, ketamine 10%; Alfasan Nederland B.V., Woerden, NL, 100 mg/mL, 10 mg/kg) and medetomidine (medetomidine hydrochloride, Sedastart; AST Farma B.V., Oudewater, The Netherlands, 1 mg/mL, 0.05 mg/kg) as induction of the anesthesia. Subsequently, after taking their body weight, the macaques were transferred by an isolator transport car, to the scanner room.

After arrival in the scanner room, artificial tears were applied to prevent the cornea from getting dry. Using a laryngoscope, the epiglottis and vocal cords were anesthetized by spraying with Xylocaine 0.1% spray (Xylocaine^®^ 100 mg/mL spray, Aspen Pharma Trading Ltd., Dublin, Ireland). Subsequently, an endotracheal tube was inserted into the trachea and inflated, and fixated to the head using a bandage. Anesthesia was maintained with the inhalational anesthetic isoflurane. The endotracheal tube simultaneously facilitated mechanical ventilation. An intravenous (IV) catheter was inserted in the left or right cephalic vein, which was connected to a GE Healthcare injection set through a three-way venous cannula. A drop of blood from the venous catheter was collected to measure blood glucose. The animal was transferred from the preparation table to the PET–CT bed, which was covered with absorbing material. The animal was secured using vet-rap tape (strap band, 5 cm × 4.5 m, Genia, St. Hilaire de Chaléons, France) in a dorsal position, head first towards the PET–CT and arms stretched upwards, and making sure not to compromise airways and respiratory movements. A rectal temperature probe was inserted anally and an oxygen saturation measurement probe was applied to the animals’ toes. Body temperature of the animal was maintained by using the Bair Hugger (3M™, St. Paul, MN, USA) supplied with 43 °C air flow. For anesthetic maintenance, a minimum alveolar concentration of isoflurane (iso-MAC) of around 0.80–1.00% was used. The respiratory parameters were set at: respiratory rate 15/min, inspiration:experiation ratio (I:E ratio) 1:4, Pmax (maximum pressure) 17, the tidal volume 10–15 mL/kg and the positive end-expiratory pressure (PEEP) 4. Oxygen- and air-flow rate were both set at 0.70 L/min. The breath-hold was induced at maximum inspiration by manually blocking the airflow for the entire scan period (approximately 35 s). Syringes with the PET-tracer were ordered and standardized to specific time points rather than individual animal characteristics. Typically, around 100 MBq of ^18^F-FDG in 4 mL was applied IV (GE Healthcare, Leiderdorp, The Netherlands) and then flushed with 10 mL 0.9% NaCl. After completing the scanning procedure, isoflurane was switched off, the respiratory rate and the tidal volume were reduced to induce an increase in the percentage CO_2_, which initiated spontaneous breathing. After a few minutes, when spontaneous breathing recurred, the setting was switched to free ventilation. After the scan, upon return to their home cage, atipamezole hydrochloride (Sedastop, ASTFarma B.V., Oudewater, The Netherlands, 5 mg/mL, 0.25 mg/kg) was administrated IM to antagonize medetomidine.

## 3. Results and Discussion

We discuss the PET–CT protocol in the four main parts: animal characteristics and preparation, scan acquisition, reconstruction, and data analysis.

### 3.1. Animal Characteristics & Preparation

Macaques have species-specific body size ranges and individual variation, both relevant to consider, as body mass and composition impact on camera settings and outcome measures. While this work relates to rhesus macaques and cynomolgus macaques only, the considerations and concepts are generically applicable and relevant for all NHP PET–CT approaches [8,13]. The points integrated in this section are body composition, blood glucose level, anesthesia, and animal positioning. 

#### 3.1.1. Body Composition

In the experimental design for NHP studies, inclusion and exclusion criteria are defined for animal selection. Next to genotype, gender, age, and indicators for social housing, bodyweight is often used as a selection criterium. The rationale for these criteria are study specific. For PET–CT purposes in particular, body weight is a relevant parameter to consider as an animal selection criterium. Imaging fat(ter) animals comes with a compromise and an upper limit may need to be set to avoid loss of image quality by beam attenuation. Beam attenuation is the absorption of X-rays that is negatively impacted by body fat, as a result of which less X-rays will reach the detector. Body fat typically (but not necessarily) increases as body weight goes up and, as this implies that X-rays reaching the detector are decreasing, it results in loss of image quality. This is illustrated in Figure 2 by showing the coronal slices of thorax CTs from three male rhesus macaques over a range of body weights [14]. A similar phenomenon occurs with PET: an increase in weight results in more photon attenuation and scattering. Thus, to preserve image quality, it is important to consider the weight in the animal selection process, and obese macaques are preferentially not included in a PET–CT study. 

Notwithstanding the former, weight is a one-dimensional quantity and only partly informing about body composition and the amount of fat. A weight-for-height index (WHI) to represent the body and fat composition is more accurate. For a WHI, the body weight is divided by height to a power X and this power X is species-specific. A power X of 3.0 for rhesus macaques and 2.7 for long-tailed macaques has been found to most optimally represent adiposity in these species [15]. The relative adiposity range for normal weight by WHI was set at 42–67 kg/m^2.7^ for rhesus macaques and 39–62 kg/m^3^ for long-tailed macaques [15]. In Figure 2, scans are depicted of three male rhesus macaques over a range of increasing body weight and WHI from left to right. In the left panel, the quality of the scan appears fine. The scan in the middle is still of acceptable quality although noise is increased and contrast resolution is decreased. For the animal in the right panel the image quality is further decreased to such a level that the likelihood of missing lesions in diagnostic assessments (at least in experimental setting) becomes unacceptably high. Based on the WHI ranges in relation to image quality, we defined an upper WHI limit of 65–70 kg/m^3^ for both macaque species. Dependent on scanner type, an additional selection parameter is defined by the inner bore diameter of the camera. For the LFER scanner the inner bore diameter is 26 cm. Therefore, we specify, equally important to WHI range, a maximal abdominal circumference of 60 cm [15]. 

#### 3.1.2. Blood Glucose Level

Both FDG and natural glucose are taken up via the same glucose transporters (GLUTs) that are expressed on the surface of cells. Upon GLUT binding, FDG/glucose is shuttled into the first step of the glycolytic pathway. In contrast to natural glucose, ^18^F-FDG becomes irreversibly entrapped at this first step of the glycolytic pathway. If high levels of endogenous glucose are available, there will be competition between natural glucose and ^18^F-FDG to be transported, which impacts on the distribution of tracer and the target-to-background ratio of the PET signal [16]. Thus, for an optimal body distribution of ^18^F-FDG, it is required to minimize the amount of endogenous glucose in the blood. This is typically achieved by fasting animals prior to PET. After fasting, the blood glucose level (BGL) should be below 7 mmol/L or 126 mg/dL, which is well within the limits recommended by the European Guidelines of Nuclear Medicine of 11 mmol/L or 200 mg/dL [16,17]. By our experience of blood glucose measurement immediately prior to FDG tracer injection, an average blood glucose level of 5.6 mmol/L (sd 1.2, range 3.0–9.2, *n* = 460) can be obtained by overnight fasting. Within this range, we can expect that competition and interference of natural glucose with FDG is minimal [4,16,18]. 

#### 3.1.3. Anesthesia

Motion artefacts compromise image quality. Therefore, NHPs are anesthetized for PET–CT acquisitions [19]. Besides anesthesia, mechanical breathing is enforced upon NHPs, not only to warrant image resolution, but also animal condition over a longer period of time. While a PET–CT procedure typically takes about 1.5 h, mechanical ventilation is recommended for maintenance under anesthetics to optimally control breathing, preserving the pCO_2_ value between 4.5% and 5.5% and heart rate at 70 to 110 bpm. Besides, the As Low As Reasonable Achievable (ALARA) guidelines for radiation recommend to keep distance from the radioactive animal as much as possible, which is more challenging when using injection anesthetics only, as the stability of the animal is to be checked regularly manually. Anesthesia is typically induced by sedation with an intramuscular injection of ketamine (10 mg/kg) in combination with the α2-adrenoceptor agonist medetomidine hydrochloride (0.05 mg/kg). The addition of an α2-adrenoceptor agonist to ketamine is absolutely required when mechanical ventilation is applied, since ketamine on its own fails to induce a significant depression of ventilation that is a prerequisite for forced breathing [20,21,22]. 

For anesthetic maintenance, an alveolar concentration of isoflurane (iso-MAC) of around 0.80% is already sufficient as the PET–CT is a minimal-invasive procedure. After intubation of the anesthetized animal, there are two main options for mechanical ventilation, by Volume Controlled Ventilation (VCV) or by Pressure Controlled Ventilation (PCV). The main difference between the two is that with VCV the tidal volume is set and kept constant while the maximum pressure (Pmax) can be variable, whereas with PCV, the peak inspiratory pressure is kept constant and, thus, tidal volume varies over time. Tidal volume is the volume of air delivered to the lungs with each breath (read: stroke of the mechanical ventilator). VCV is preferred above PCV as a constant tidal volume over the full timeframe of scanning warrants stable images and, importantly, spatial reproducibility. A disadvantage of VCV may be the induction of lung injury when the airway pressure increases due to a (pathological) obstruction (as the mechanical ventilator is forced to a particular volume). Such an obstruction, for instance, can occur due to an infectious respiratory disease, causing lymph nodes in the mediastinum to be enlarged. To circumvent VCV-associated lung injury it is essential to control several parameters [23]. One of those parameters is the time ratio of inspiration over expiration (I:E ratio). The normal I:E ratio at rest and while asleep is 1:2 or less. Inspiration normally is an active process, while expiration is passive, resulting in a longer silent phase. The PET–CT imaging acquisition protocol should balance between animal health and acquiring high quality images. With VCV, to avoid air-trapping (breath stacking) and an increase in airway pressure due to an obstruction, the ratio is set at 1:4 at a respiratory rate of 15/min. To prevent lung damage from a high inspiratory pressure, Pmax needs to be set at 17. To supply enough oxygen to the body, tidal volume needs to be set at 10–15 mL/kg. Oxygen- and air-flow rate are both set at 0.70 L/min to ensure enough isoflurane reaches the macaque. Finally, the Positive end-expiratory pressure (PEEP) is a critical parameter. PEEP during mechanical ventilation refers to the pressure that is set at the end of each ventilation cycle to avoid the collapse of alveoli. Maintaining a PEEP of 4 cm H_2_O has proven adequate to prevent such collapse. All parameter settings are summarized in Table 1. 

After completing the scanning procedure, the respiratory rate and the tidal volume need to be reduced, to 6/min and 40 mL respectively, to induce an increase in the percentage CO_2_ and therefore initiate spontaneous breathing and recovery. After several minutes, when the animal resumes spontaneous breathing, mechanical ventilation can be switched off.

When the animal has returned to its home cage, atipamezole is provided (0.25 mg/kg). Atipamezole is the antagonist of medetomidine leading to the quick recovery of normal function [20,21,22].

#### 3.1.4. Animal Positioning

The positioning of macaques and other similar sized NHPs in the scanner is relevant for diminishing scan artefacts and maintaining body temperature. For human whole-body PET–CT scans, guidelines have been published for the position of the arms. Putting the arms besides the body may lead to artifacts due to shadowing of the arms [4,17,24]. Those artifacts can occur in scanning the torso of macaques too. For this reason, it is recommended to stretch the arms upwards in macaques, similar to what is commonly done in human scanning. 

Maintaining body temperature is key for optimal scan results without interference of brown adipose tissue (BAT) activation. BAT is involved in temperature regulation, activated in cold environments to maintain the body temperature by producing heat through glucose metabolism. This can lead to an increased ^18^F-FDG uptake in BAT. In both human and NHPs, BAT is predominantly found, in both human and NHPs, in the cervical, supraclavicular and thoracic paravertebral region by which it can interfere with the scan results of the lung [25]. 

To prevent BAT activation the macaques are placed on a heating pad during preparation. Although the chance of BAT activation is found to be less than 5% in humans, this percentage might be higher for macaques. In case of an unnoticed breakdown of our heating pad, we found interfering BAT activation in 4 out of 4 animals, as illustrated in Figure 3 [26,27]. 

After animal preparation, a convective temperature management system, the 3M™ Bair Hugger™ (3M™, St. Paul, MN, USA), is installed to keep the animal on temperature during the scan. The 3M™ Bair Hugger™ uses a forced air system to keep the NHP warm without interfering with the scan. We placed the 3M™ Bair Hugger™ at the rear end of the bore, and, therefore, the most beneficial approach is to place the NHP head-first-supine (HFS) in the scanner (Figure 4). In this position, nearly the entire body is within the bore of the scanner. 

### 3.2. Scan Acquisition

The acquisition of the scan is at the center of the entire procedure in which a CT and PET are acquired subsequently. In combined imaging strategies with PET, the CT is used to obtain anatomical information, serving as a reference for positioning the PET field of view (FOV) and for attenuation correction, which is correcting the PET signal for differences in photon absorption.

#### 3.2.1. CT

For a thorax PET–CT, a non-contrast CT is necessary for attenuation correction and sufficient for adequate discrimination of the lung lesions within the lung parenchyma. Nonetheless, the use of an IV CT contrast agent may add by improving lesion detection and characterization of particular parts of the thorax. With a contrast agent the x-rays are changed due to absorption and/or scattering. Iodine is used as element to realize both scatter, by an increased density, and absorption. Different concentrations of iodine are used in the clinical available agents. However, the appearance will mostly be dependent on the timing of the injection and image acquisition [28].

For vascular processes within the mediastinum, the application of an intravenous contrast agent is recommended to improve the sensitivity of CT towards recording abnormalities [29]. Whether to use a CT contrast agent or not should not only be based on the target tissue and the biological process of interest. It should also be driven by considerations, such as possible interference of contrast in sequential CT recordings or the risk on adverse side-effects, such as nephropathy that has been associated with the application of contrast agents [30]. Although rare at the concentrations typically recommended, under preclinical research conditions of experimental disease and/or treatment, synergistic interference can have an unwanted impact on kidney function [29].

Three contrast application regimes were performed to determine how those would improve the visualization of vascular processes in the mediastinum (Figure 5). For all three regimens, the contrast (Omnipaque 300 mg/mL, GE Healthcare, Chicago, IL, USA) was injected IV in a timeframe of 10 s followed by IV flushing with 0.9% NaCl for 20 s. Regime were as follows: (1) flush with 40 mL, start scan 120 s post injection, (2) flush with 40 mL, start scan 70 s post injection, and (3) flush with 10 mL, start scan 30–40 s post injection. With the third regime smaller vessels became detectable, which stayed undetectable without the use of a contrast agent and in the first two regimes. However, the additional value of a contrast CT for the thorax stays mostly limited for macaques and must always be carefully weighed against the risk of possible adverse effects and additional CT, which need to be obtained. 

In man, typically, a whole-body CT can be performed in only a couple of seconds, by which the impact of breathing artefacts is neglectable. With the LFER 150, however, this is different. To obtain high-resolution images by small isotropic voxels with a resolution of 200 micron the LFER uses a cone beam CT (CBCT). The scanning time of a CBCT on the LFER is 35 s, making respiratory control to avoid breathing artefacts a prerequisite. There are multiple ways to achieve respiratory control, and one of the most commonly used is inducing a breath-hold on maximum inspiration by manual interference on the mechanical ventilation [31]. It is beyond the scope of this paper to discuss this further (but under consideration for publication elsewhere; Tolgyesi et al. submitted 2020).

#### 3.2.2. PET Circulation Time

After injection, the pharmacokinetic profile of ^18^F-FDG in man flattens in about 60–90 min [32]. Based on that profile, a circulation time of 60 min is recommended before the start of a clinical ^18^F-FDG PET [17,33]. The pharmacokinetic profile and the circulation time, however, are dependent on the species-specific basal metabolic rate (BMR) as well as the individual body weight [34,35]. The higher the BMR, the shorter the circulation time should be; and the higher the weight, the longer the circulation time. The BMR for man is almost similar to that of macaques: 1.50 and 1.55, respectively [36]. The weight, and body surface differ and is around 10 times smaller for macaques. Due to this size difference, in theory, the ^18^F-FDG circulation time for macaques is substantially shorter. However, there is a third parameter involved, which is the scanner. Based on the LFER and analyses regarding the scanning time and activity, we determined that a circulation time of 45 min is recommended before starting a thorax PET in NHPs. 

#### 3.2.3. PET Scanning Time & Activity

PET image quality, based on the signal to noise ratio (SNR), is negatively influenced by higher body weights, as explained in the body composition paragraph. An increase in weight leads to more photon attenuation and a higher scatter fraction. This phenomenon can be compensated by increasing the scan time or by increasing the amount of radioactivity injected. The relation between activity and scan time is linear. Doubling the scan time allows to apply half of the amount of radioactivity. The recommended activity of ^18^F-FDG to inject is provided by the European Association of Nuclear Medicine (EANM), as depicted in Table 2 and based on body weight and a scan time of 2–3 min for each FOV. These recommendations, however, need verification or optimization towards the characteristics of the scanner in place. Two important parameters in this regard are the noise equivalent count (NEC) and the dimension of the crystals.

The NEC is a way to estimate raw data SNR and an indirect measure to image quality in combination with other parameters. A high NEC represents a favorable ratio of “true” coincident events over scattered and accidental coincidence events [37]. However, there is a high variance of NECs found in both clinical and pre-clinical scanners. Lower NECs can be compensated by increasing the scanning time [38,39]. 

With regard to crystal size: smaller crystals contribute to an increase in resolution, but more overall counts are required to achieve an appreciable level of SNR, as the amount of “true” counts per crystal is decreasing while the amount of noise stays the same. Thus, with smaller crystals, a longer scanning time is necessary to achieve similar results as obtained with larger crystals over a shorter scanning time.

In aggregate, and by our experience with the LFER, we standardized the amount of activity injected at (about) 100 MBq. This is at the high end of what strictly would be needed for good quality scanning, but it provides sufficient margin for unforeseen cases of delay in animal handling, avoiding cancellation, or compromised image quality. 

To assess the optimal scanning time with 100 MBq of radioactivity and 45 min circulation time before starting a scan, we made several reconstructions. In Figure 6, it shows that the SNR is improved for longer scan times, with improved specificity to discriminate abnormalities (arrows). Based on those reconstructions we concluded that a scan time of 15 min for each FOV is enough to reach sufficient image quality.

### 3.3. Reconstruction

An advantage of PET is the ability to quantify the data for diagnosis, in support of prognosis and therapeutic response monitoring purposes. However, PET quantification is affected by several factors. Image reconstruction parameters can have an impact of up to 30% on the standard uptake values (SUVs) in ^18^F-FDG scans. One of the other key factors that impact the quantification of PET scans, is the attenuation correction [32,40,41]. 

#### 3.3.1. Reconstruction Settings

There are various options for reconstruction settings and methods and these differ for each scanner type. Hard guidelines for obtaining comparable SUVs are not available. Nevertheless, there are some relevant recommendations to make about what to include in a standardized reconstruction algorithm to obtain a quantitative ^18^F-FDG PET image. Obligatory are, besides attenuation correction, corrections for normalization, system dead time, random coincidences, scatter and sampling non-uniformity [32,42]. In addition, for comparison of scans and SUVs, a similar image resolution is essential [32]. 

#### 3.3.2. Attenuation Correction

During PET the photons reach the detector from different places out of the body. To relate those back to the point of origin and allow quantification of the photons, attenuation correction is necessary. When using PET–CT, the attenuation correction is based on the CT, which provides the benefit of increased accuracy and the decrease in overall scanning time as compared to approaches based on a transmission scanning with a rod source. Attenuation correction is applied by creating a material map from the Hounsfield Units (HU), generated by the CT at 80 keV and transformed into the correct energy of 511 keV [43,44]. The material map is not an exact copy of the CT as the range of HU is reduced to a limited number of intervals that relate to different tissues and their specific X-ray absorbing potentials. The more intervals representing types of tissues, the more accurate the attenuation correction. However, distinguishing more tissue types is only effective if the fusion of the CT and PET is optimal. In this regard, while CTs are obtained during a selected period of the breathing cycle, for PETs, this is impossible as data acquisition takes more time. This results in slight misrepresentations of certain parts of the scan due to natural movement. Especially in the lower lobes of the lungs, those differences can become prominent and compromising to the quality of CT and PET fusion. Therefore, a more robust representation of the lung versus body density will benefit from limiting the amount of tissue types for creating a material map. The influence of attenuation correction is illustrated in Figure 7A,B, in which the reconstruction of a scan without and one with attenuation correction is visualized. 

For the LFER the material map can be divided in six different ranges—air, lung, muscle, adipose tissue, and bone. This will lead to a reliable representation of the thorax. For clinical scanners this point does not need to be considered as there is a direct bilinear relation between the CT intensity in HUs and linear attenuation coefficient at 511 keV, by which all densities are represented instead of being divided in a predefined amount of ranges. 

### 3.4. Data Analysis

Extracting relevant and meaningful data from the scans is the last step of the entire PET–CT procedure. Various approaches to identify the target are available and there are several options of output parameters to choose from. 

To this end, and as the combination of PET and CT reveals functional abnormality associated with anatomy and pathology that allows for quantification, the identification of the region of interest (ROI) is a critical step in the process. Below we exemplify how target identification approaches, and choosing from various output parameter options can valuably add to PET–CT quantification of disease. 

#### 3.4.1. ROI Definition

The way to set the ROI is partly defined by the region of the body under investigation. Here we focus on thorax imaging and discuss two methods to generate a ROI in a standardized manner. First, in the lungs, lesions can be easily distinguished from surrounding lung tissue and it is advised to use scanner defined HU ranges from anatomical CT to discriminate healthy from non-healthy tissue. Subsequently, the CT-defined region can be copied to the fused, functional PET image. With this approach, an objective definition of a ROI is warranted which supports standardization of the procedure and generation of output parameters from both CT and PET. 

In the mediastinum, where it is challenging to discriminate lesions from the surrounding tissue, another approach needs to be used. Here, a lower threshold SUV can be used to differentiate between lesions and healthy tissue. The exact cut-off value is not essential as long as it is used consistently. This approach of defining a ROI, like the former one, gives output parameters with consistent reliability.

#### 3.4.2. Anatomical and Functional Results

In clinical practice, SUVmax is the most commonly used output parameter. It reflects the value of one single voxel with the highest uptake value in any particular ROI. While SUVmax is informative, there are other parameters that also provide relevant information and should be considered for result characterization of PET–CT [41].

The SUVmean is calculated as the average intensity within a ROI. The SUVpeak represents the average activity of a defined number of voxels in the peak area, and thus, is less affected by noise than the former SUVmax. This makes the SUVpeak more robust than SUVmax in particular when reconstructing with smaller voxels. The extent of pathological manifestation is reflected by the volume of the ROI, which is principally derived from the anatomical imaging modality by CT. Lastly, the SUVtotal is the total SUV of all separate voxels in a ROI. It is evident that SUVtotal is affected by manual identification of the ROI and, thus, prone to subjectivity, which can become a pertinent issue in comparative research approaches.

In Figure 8, different manifestations of a tuberculosis lesion, as example of a bacterial infection, and a SARS-CoV-2 related lesion, as example of a viral infection in the lungs of macaques, are visualized. Those scans clearly show that, in the upper row the lesions have a different appearance but in the second row the appearance is similar. The goal of data quantification is to discriminate these differences and similarities. For both examples, the aforementioned output parameters are compared for the two images that are visualized. In the upper panels, the examples show an SUVtotal that is, for both scans, roughly the same, but the SUVmean (1.3 vs. 0.8) and the volume (3086 mm^3^ vs. 4775 mm^3^) are different. This means that the total activity of the lesion is almost the same, but the percentage of the lung affected and the metabolic activity of the lesion are not. In the lower panels of Figure 8, the example shows that although the appearance is similar there are differences found for all quantification parameters.

Every parameter comes with advantages and disadvantages. Not any single output parameter is able to fully capture the manifestation of disease in the ROI as recorded by PET–CT. However, when using multiple complementary parameters, the apparent differences between lesions can be enumerated [41,45].

## 4. Conclusions

In conclusion, thorax PET–CTs in NHPs are of additional value in a range of experimental respiratory infection studies [31,46,47,48]. One of the advantages is the clinical translation ability of those scans. However, a one-to-one translation of clinical protocols and procedures is not feasible to obtain consistency in PET–CTs image quality of NHPs. Various aspects are discussed extensively in this paper, and based on this, practitioners should consider our recommendations when performing a thorax PET–CT of NHPs.

### 4.1. Animal Characteristics and Preparation

Body composition: use a WHI instead of body weight solely as NHP study selection criterion.

Blood glucose: when fasting the animals overnight prior to injection of ^18^F-FDG there is minimal competition with natural glucose expected.

Anesthesia: use a combination of ketamine and an α2-adrenoceptor agonist for induction anesthesia followed by VCV with isoflurane for maintenance. This is most stable for the animal and follows the ALARA guidelines for radiation for caretakers. 

Animal positioning: prevent BAT activation by placing the animal on a heating pad during ^18^F-FDG incubation and position them in the scanner similar to man, on the back with the arms up.

### 4.2. Scan Acquisition

CT: a non-contrast CT alone is sufficient both for anatomical information and attenuation correction.

PET circulation time before start: depends on BMR, bodyweight and the scanner, we recommend a waiting time of 45 min for thorax PET–CTs.

PET scanning time & activity: image quality is based on the SNR influenced by body weight, the NEC of a scanner, amount of radioactivity and scanning time. We advise, with 100 MBq and 45 circulation time a scan time of 15 min for a single thorax FOV.

### 4.3. Reconstruction

Reconstruction parameters: those settings differ between scanner types, but obligatory are corrections for normalization, system dead time, random coincidences, scatter and sampling non-uniformity.

Attenuation correction: attenuation correction is necessary to be able to quantify the data. Optional is the choice in the level of complexity, by adding more tissue types, which will decrease the robustness of the reconstructed scan. We suggest a complexity of six different tissue types.

### 4.4. Data Analysis

ROI definition: a ROI could be defined based on anatomical, with a HU range, or functional information, with a SUV cut-off. Key is a standardized approach to ensure objectivity and consistent reliability.

Anatomical and functional results: using multiple output parameters, which are complementary to each other, are recommended and will improve the similarity between the qualitative and quantitative data analysis. 

Finally, PET–CTs in general and in this regard thorax PET–CTs offers a unique and potent minimal-invasive imaging method for studying infectious respiratory diseases over time in macaques. When following the recommendations in this paper it allows reliable longitudinal and quantitative imaging to obtain more information out of any single animal. They strengthen PET–CTs as a meaningful refinement to experimental studies with macaques.

## Figures and Tables

**Figure 1 animals-11-00204-f001:**
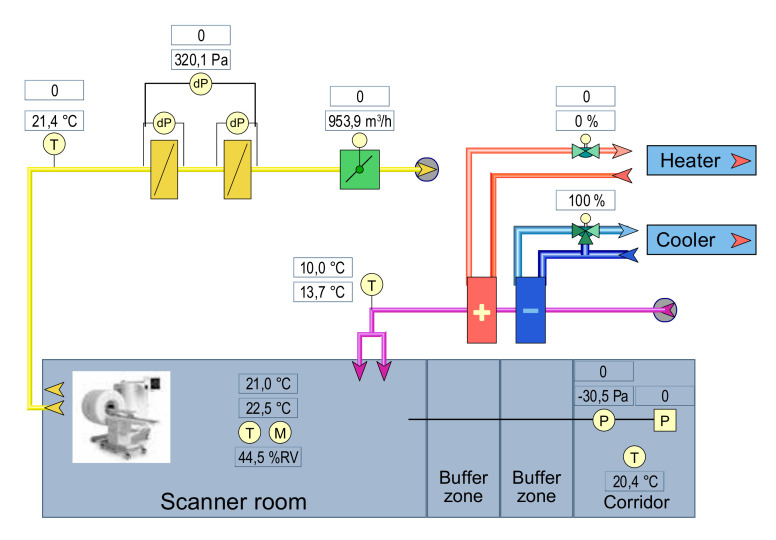
Schematic representation of the scanner room, temperature and air pressure. The temperature set to enter the room is ideally 10.0 °C and the room temperature 21.0 °C.

**Figure 2 animals-11-00204-f002:**
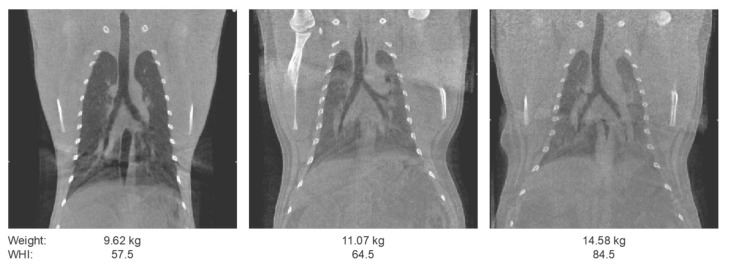
From left to right, three representative coronal sections of a thorax computed tomography (CT) are presented with their respective weight and weight-for-height index (WHI). With an increasing weight the contrast between the lungs and surrounding tissue is decreasing. While the amount of noise over the entire image is increasing. Similar window-level settings are applied to all sections.

**Figure 3 animals-11-00204-f003:**
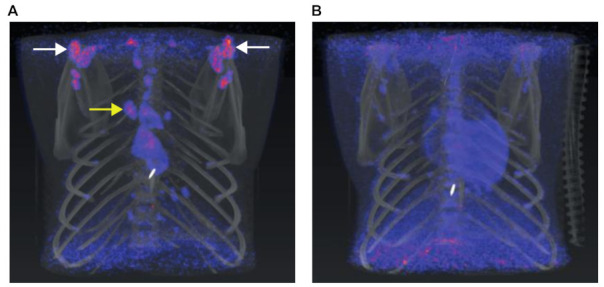
(**A**) Brown adipose tissue (BAT) activation is most prominent around the supraclavicular (white arrows) and paravertebral (yellow arrow) region. (**B**) The same animal scanned at another timepoint without BAT uptake. Similar window-level settings are applied to both scans.

**Figure 4 animals-11-00204-f004:**
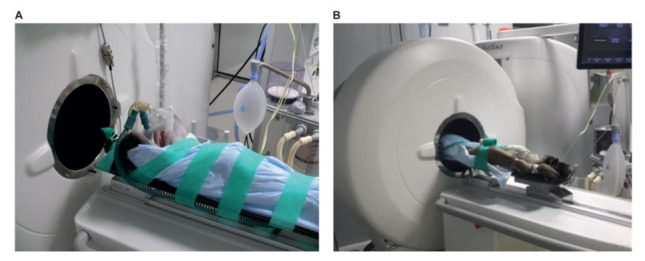
Visualization of the position of the animal on the scanner bed (**A**) and while scanning (**B**).

**Figure 5 animals-11-00204-f005:**
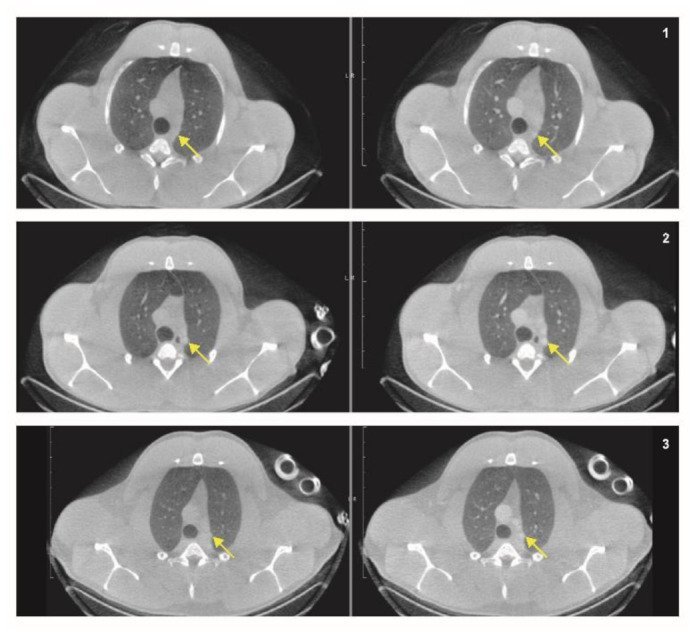
A representative pre-contrast (left) and post-contrast (right) transversal CT-slice are displayed of three different contrast regimes. For all three the contrast is injected intravenously (IV) in a timeframe of 10 s followed by IV flushing with 0.9% NaCl for 20 s. 1. IV flush with 40 mL, start scan 120 s post injection, 2. IV flush with 40 mL, start scan 70 s post injection, 3. IV flush with 10 mL, start scan 30–40 s post injection. The arrows are pointing towards the increased visualization of the vessels with the different contrast application regimes. Similar window-level settings are applied to all CT-slices.

**Figure 6 animals-11-00204-f006:**
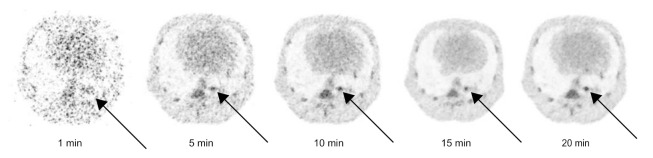
PET representations of multiple reconstruction time frames of the same scan visualizing the increase in signal to noise ratio (SNR) while increasing the scanning time. The black arrow points towards an abnormality. Similar window-level settings are applied to all reconstructions.

**Figure 7 animals-11-00204-f007:**
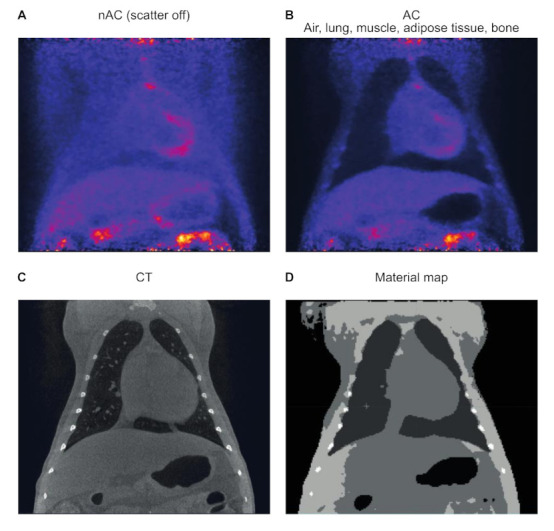
Representation of the same scan and cross section without (**A**) and with (**B**) attenuation correction, including the corresponding CT (**C**) and representative material map in which the different ranges used to discriminate different tissue densities (**D**). The attenuation correction is based on six different ranges: air, lung, muscle, adipose tissue, and bone. The lungs are only discriminated with attenuation. Similar window-level settings are applied to all cross sections.

**Figure 8 animals-11-00204-f008:**
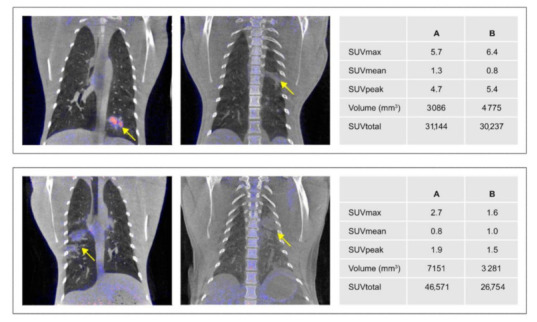
Two rows of images, each illustrating two different manifestations (A and B) of a bacterial infection (upper row) and a viral infection (bottom row). The respective lesions are indicated by the yellow arrows. Representative output parameters are shown on the right for the left (A) and the right (B) images, respectively. Similar window-level settings are applied to all images.

**Table 1 animals-11-00204-t001:** Overview of all major respiratory setting during volume controlled mechanical ventilation (I:E = inspiration:expiration, PEEP = positive end-expiratory pressure).

Parameter	Value
I:E ratio	1:4
PEEP	4 cm H_2_O
Oxygen flow rate	0.7 L/min
Air flow rate	0.7 L/min
Pmax	17
Tidal volume	10–15 mL/kg
Respiratory rate	15/min

**Table 2 animals-11-00204-t002:** Recommended ^18^F-FDG activity, for men, for a positron emission tomography (PET) of the torso.

Weight (kg)	Activity (MBq)
3	26
4	30
6	44
8	55
10	70
12	81
14	92

## Data Availability

No new data were created or analyzed in this study. Data sharing is not applicable to this article.

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
