# Peer review of "Recommendations for Standardizing Thorax PET–CT in Non-Human Primates by Recent Experience from Macaque Studies"

_animals, 2021, doi:10.3390/ani11010204_

Round 1
Reviewer 1 Report
My review is limited to the physics of PET-CT. I have only minor comments that can be easily addressed.
- In several parts of the manuscript (page 1, line 17; page 2, line 65), the authors use the words "contrast agent" for the PET radiotracer. Whereas it is appropriate for CT and MRI, it sounds strange for PET. A radiotracer like FDG is a probe that can be better qualified as a molecular imaging agent.
- Page 1, line 16: I do not understand what is meant by "dynamics" in "distribution dynamics of a specific contrast agent". Do you mean the pharmacokinetics of the radiotracer (dynamic imaging)? In your study, you perform static imaging (wait until an equilibrium is reached)
- Subsection 2.2.
-
- “The scanner operates at submillimetre resolution…”. Do you refer to the CT part or both parts? PET resolution is not submillimetre.
- For PET, it is better to also provide some quantitative numbers for its intrinsic performances, like the effective spatial resolution across the field-of-view and the sensitivity. You can quote/cite the following publication: https://doi.org/10.2967/jnumed.117.206243.
- It is better to report the size of the crystal section as 1.51 mm x 1.51 mm (might not be isotropic).
- The high resolution by itself is not due to the small gantry bore (increases the depth-of-interaction effect) or to the size of the FOV. It is linked to the crystal size (plus other factors, like light collection).
- Subsection 3.1.1, page 5, line 179. The intensity of the X-ray (mAs) can be modulated to compensate for high WHI, like for humans. Did you consider it, or a flat mAs value was used, regardless of the morphology of the NHP?
- Page 5, line 182: “showing the projection of thorax CT”? It’s not a coronal slice?
- Page 5, Figure 2. The images contain quite some artefacts, even at low WHI. Do you also suffer from beam hardening?
- Figures 2, 3, 5, 6, 7 and 8. I recommend to attach the colour scale, both for CT in HU (its choice is crucial) and PET in SUV.
- Subsection 3.2, page 8, line 301. Attenuation correction in PET is crucial even if the tissues densities are uniform. It corrects for photon absorption, not for non-uniform tissue densities.
- Figure 5. The visual comparative interpretation of the slices is not easy on a PDF. Could you help the reader, by either adjusting the limits of the colour scale, or adding arrows to guide its eyes?
- Subsection 3.2.1, page 9. For humans, it is common practice to gate chest PET (and possibly CT) data to account for respiratory motion. Why this option was not considered in this study?
- Subsection 3.2.3, page 10, line 361. The Noise Equivalent Count (NEC) is a way to estimate raw data SNR. Its link to reconstructed image quality is no direct; the latter depends on many other parameters. Setting the activity to be at the peak NECR does not necessary correspond to an optimal image quality. It might be better to increase the scan duration rather than the injected activity.
- Table 3. It looks like a black box that can only be understood by the users of the same system. Could you elaborate a little more and avoid acronyms without explanation, like Tera-Tomo 3D. In addition, do not forget the units for the voxel size.
- Subsection 3.3.2. Do you have access to the PET mumaps? If yes, it’s worth displaying them. Could you explain why adding more tissue types decreases the robustness of the reconstructed PET image. For human PET/CT systems, there is a direct bilinear relation between the CT intensity in H.U. and the linear attenuation coefficient at 511 keV. Why such a relation is not appropriate for the NHP PET-CT system?
- Figure 8. Can you show to what correspond lesions A & B? I clearly see the lesion in the first column (A), but not in the middle column (B ?).
Reviewer 2 Report
Review Animals Stammes et al
The manuscript by Stammes et al does a good job of describing the experimental protocol used by their laboratory in imaging chest PET/CT in NHPs. The paper is clearly written, although editing by a native English speaker is needed. Additionally the manuscript would benefit from a broader description of other pre-clinical PET/CT systems that can be used to image NHPs, as well as clinical PET/CT scanners and other useful tracers in addition to FDG.
- Overall the manuscript needs editing from a native English speaker for syntax and grammar
- ‘For functional molecular imaging the use of a specific contrast agent is required to be able to assess a biological process through a dedicated molecular target[3].’ : in the context of PET imaging it is not appropriate to refer to contrast agents, but rather tracers
- Figures would benefit from more detailed figure legends and some more details on the figures themselves. For example in Figure 3, the location and FDG uptake of BAT should be pointed out for clarity
- Figures should be higher quality (figure 4 is very low quality)
- In the section describing the use of CT contrast, the authors fail to mention which kind of contrast agents can be used, their differences and different usages
- Instead of PET incubation time, it would be better to use the more common phrase ‘circulation time’
- Figure 6: ‘red arrow’ should be ‘black arrow’
- Figure 8: arrows or insert panels should be used to clearly mark the lesions
- Many considerations are specific to the pre-clinical PET/CT system used by the authors: it would be nice to have a paragraph describing other available systems for imaging NHPs, with a description of differences, and also a description of potentially different procedures when clinical PET/CT systems are used
- More examples of other pathologies should be given in the figures
- It would be nice if the authors would discuss other potentially useful tracers in addition to FDG
